# INCYDE: A LARGE SCALE CYCLONE DETECTION AND INTENSITY ESTIMATION DATASET USING SATELLITE INFRARED IMAGERY

## ABSTRACT

Tropical cyclones are devastating natural phenomena that cause a significant amount of damage every year. Conventionally, the Dvorak technique is used to manually estimate cyclone intensity from satellite infrared imagery by following a set of rules to identify certain cloud features. The manual nature of the Dvorak technique introduces subjectivity, necessitating the implementation of an automated process for cyclone intensity estimation. Satellite infrared imagery provides valuable information for detecting cyclonic storms. Recently, deep CNN models have proven to be highly efficient in detecting relevant patterns in the images. In this work, a novel cyclone detection and intensity estimation dataset called INCYDE (INSAT-based Cyclone Detection and Intensity Estimation) dataset is presented. The cyclone images in the dataset are captured from INSAT 3D/3DR satellites over the Indian Ocean. The proposed INCYDE dataset contains over 21k cyclone images taken from cyclones over the Indian Ocean from the year 2013 to 2021. The dataset pertains to two specific tasks: cyclone detection as an object detection task, and intensity estimation as a regression task. In addition to the dataset, this study introduces baseline models that were trained on the newly presented dataset. The results of this research would help develop innovative cyclone detection and intensity estimation models, which in turn could help save lives.

## 1 INTRODUCTION

Tropical Cyclones (TC) are highly destructive natural phenomena and they are one of the costliest natural disasters that cause a wide range of hazards. Tropical cyclones form over warm ocean waters when the water temperature is at least 26.5 degrees Celsius (80 degrees Fahrenheit). As the storm forms, it begins to suck up more and more warm air and moisture from the ocean, which causes it to grow larger and more powerful.

Deep learning has proven to be very efficient in detecting fine patterns in images, the cloud patterns in satellite infrared imagery provide valuable information about the formation of a cyclone. Thus, the usage of deep learning techniques to detect and estimate the intensity of cyclones using infrared satellite images has a lot of potential to be used as an early warning system to predict cyclones. Thus, there is a need to have a large-scale dataset for cyclone detection and intensity estimation to help research and develop deep Convolutional Neural Network (CNN) models that can help provide early warning for cyclones using IR satellite imagery without any human intervention. In literature, there has been some well-documented datasets for cyclone intensity estimation, but the present datasets lack cyclone detection capabilities and lack some geographical areas like Indian Ocean.

In this work, we propose a large-scale dataset called INCYDE dataset for cyclone intensity estimation and cyclone detection. The dataset includes infrared (IR) images from INSAT 3D/3DR Mahammad; Kaushik satellites from year 2013 to 2021 annotated using the Indian Meteorological Department's (IMD) best track data IMD (2022). The proposed dataset has higher-resolution images and covers a new geographical region (Indian Ocean Region) which is not present in other datasets in the literature. Additionally, the proposed dataset also introduces the research community to a new task of cyclone detection in satellite infrared images. The INCYDE dataset is available at https://doi.org/10.5281/zenodo.8015544

To summarise, the contributions of this work are as follows:

- A large-scale dataset INCYDE for cyclone detection and intensity estimation is presented.
- The dataset creation pipeline used to curate the dataset has been presented.
- A thorough dataset analysis has been carried out to provide insights relevant to the dataset and a review of other state-of-the-art (SOTA) datasets has been carried out.
- Various deep CNN models have been trained on the proposed dataset for cyclone detection, and cyclone intensity estimation to act as baseline models for future work.

The rest of the paper follows the following structure. In section 2, the literature survey is carried out where prior works in the field of cyclone intensity estimation are discussed. In section 3 the methodology used to carry out the experimentation work in this research work is presented. In section 4, the proposed INCYDE dataset is presented along with important characteristics of the dataset like sample images, dataset curation process, dataset split, and comparison with other publicly available datasets. In section 5, the experimentation regarding baseline models trained on the proposed dataset is presented.

The motivation for our paper stems from the critical need to address cyclone intensity estimation as a computer vision research problem. Currently, the field of cyclone intensity estimation heavily relies on remote sensing expertise, making it inaccessible to many computer vision researchers who could contribute valuable insights. Our aim is to bridge the gap between remote sensing and computer vision, making cyclone intensity estimation more accessible and fostering collaboration between the two domains. To accomplish this, we have curated an image dataset specifically for cyclone intensity estimation and provided an object detection task dataset in Common Object in Context (COCO) format. This research work would help researchers develop robust cyclone intensity estimation and cyclone detection frameworks that have the potential to help save countless lives by providing early warnings for cyclones.

## 2  RELATED WORK

Conventionally, the Dvorak technique is used to estimate cyclone intensity. The Dvorak technique Dvorak (1973) uses satellite images to estimate the intensity of a tropical cyclone using its cloud pattern. The cloud pattern is analyzed in terms of the distribution of cloud cover and the temperature of the cloud tops. The cyclone is classified using a combination of cloud pattern features such as the size, shape, and temperature of the cloud tops, as well as the presence of an eye or the center of the storm. Once the storm has been classified, the Dvorak technique provides an estimate of the maximum sustained winds and the central pressure of the storm. This information can be used to issue forecasts and warnings to people in the path of the storm.

While the Dvorak technique is characterized by its inconsistencies and heavily manual approach, relying on the subjectivity of experts. In contrast, the deep learning method offers a more systematic and objective alternative. Various datasets have been published in the literature for cyclone intensity estimation. Maskey et al. (2023) released a dataset for cyclone intensity estimation which includes cyclones in the Atlantic and East Pacific Oceans from the year 2000 to 2019 for the Tropical Cyclone Wind Estimation Competition. Maskey et al. (2020) also developed a simple deep CNN with 6 CNN with max-pool layers and 3 fully connected layers at the end. Their baseline model was able to achieve a root mean squared error (RMSE) of 13.24 knots. They also developed a web application to visualize historical data of cyclones and get current cyclone intensity estimation predictions. Maskey et al. (2020) used interpolation to generate more labels from existing labels. TCIR Chen et al. (2018) is another dataset for cyclone intensity estimation task, It includes 4 channels (Infrared, Water Vapor, Visible, and Microwave bands) satellite infrared images in the West Pacific, East Pacific, and Atlantic Ocean Regions, the dataset contains satellite images having 301x301 pixels size, the authors of TCIR have also employed interpolation to generate more labels from existing labels. HURSAT Knapp & Kossin (2007) is another such dataset for cyclone intensity estimation from years 1978 to 2015 having 3 hourly satellite images. HURSAT also uses interpolation to generate more labels from existing labels. The satellite images are abundant and have higher frequency than the labels for those images, thus the approach of using interpolation to generate labels between two consecutive timestamps helps ensure the efficient use of abundant satellite images.

Li & Chen (2021) presents a cyclone dataset FY4A-TC using multispectral images of 81 cyclones captured by FY4A satellite from 2018-2021. Li & Chen (2021) uses a Convolutional Neural Network (CNN) with a self-label regularizer to increase accuracy. The author proposes a generative adversarial network-convolutional neural network (GAN-CNN) hybrid model that uses both passive microwave rate(PMW) and Visible channel images. Tan et al. (2022) used Himawari-8 satellite products for cyclone intensity estimation. Tan et al. (2022) used a convolutional neural network (CNN) and they were able to achieve 4.06 m/s (7.89 knots) RMSE. Their work revealed that the model's performance is highly affected by the initial cloud products. Chen et al. (2020) presents a framework for tropical cyclone intensity estimation using a generative adversarial network (GAN) to handle temporally heterogeneous datasets. Their model uses IR1 and WV channels for prediction, eliminating the dependence on the PMW channel. They use a hybrid GAN-CNN model, with two generators for producing VIS and PMW images. Bloemendaal et al. (2020a) uses STORM dataset Bloemendaal et al. (2020b) for TC wind speed estimation. The author uses historical best track data from the International Best Track Archive for Climate Stewardship (IBTrACS20) and generates tropical cyclone data comparable to 10,000 years with the current climate constraints. The authors propose the STORM dataset and use it to find the return periods of a Tropical cyclone hazard. Chen et al. (2019) used a deep CNN model for estimating TC intensity using satellite IR brightness temperature and microwave rain rates together with additional TC information like the basin, day of the year, local time, etc. They managed to achieve an RMSE of 8.79 knots for a subset of 482 samples. They use a 4-layer CNN model with three fully connected layers with random rotation as preprocessing and post-analysis smoothing to achieve lower RMSE. Miller et al. (2017) used the Geostationary Operational Environmental Satellite (GOES) program's IR images for historical tropical storms in the Atlantic and Pacific basins from the year 2000 to 2015. Miller et al. used the HURDAT2 dataset for labels. Lu & Yu (2013) used IR satellite images from 2006 to 2010 for cyclone intensity estimation. Pradhan et al. (2018) used a deep CNN for cyclone intensity estimation. Luo et al. (2021) presented a novel DR-transformer for tropical cyclone intensity estimation. Their proposed model is able to achieve a SOTA RMSE of 7.6 knots. Their transformer-based model extracts Distance-consistency(DC) and rotation invariance(RI) features in TC images. These features extracted can overcome the issues faced by classical CNN models in differentiating highly similar visual features. Additionally, they also repurpose their model to incorporate the evolution of the cyclone through time and intensity. Various machine learning algorithms were used by Biswas et al. (2021), Devaraj et al. (2021) for predicting hurricane intensity using IR satellite imagery data. Devaraj et al. (2021) used a VGG 19 model to predict the extent of the damage.

## 3 METHODOLOGY

In this work, a novel dataset INCYDE is presented. The images in the dataset were collected from INSAT 3D and 3DR satellites Mahammad; Kaushik and the labels were collected from IMD best track data. The IMD best track contains cyclone information like latitude, longitude, maximum sustained wind speed, timestamp, etc. The data in the best track data has a frequency of 6 hours but the INSAT 3D/3DR satellite IR imagery is available in 30/15 minutes intervals. So, in order to efficiently use all the images available, it was imperative to generate more labels for the images in the dataset using the available data from IMD's best track data. So, we used interpolation to generate more labels between two consecutive timestamps in the best track data for the corresponding images. The details of the interpolation step are explained in later sections.

The proposed dataset, i.e., INCYDE contains 21k images or over 68k images with augmentation from INSAT 3D and INSAT 3DR satellite IR imagery Mahammad; Kaushik of cyclones between the years 2013-2021. Files in the INCYDE dataset are named with the cyclone date and time, followed by an indication of the augmentation, if any. This aids users in quickly identifying both augmented and original images, thus leaving the choice to researchers whether to utilize the provided augmentations or solely rely on the original images for their studies. The INSAT 3D/3DR provides IR imagery in 6 different bands (VIS (Visible) 520 - 720 nm, SWIR (Short Wave Infrared) 1550 - 1700 nm, MWIR (Mid Wave Infrared) 3800 - 4000 nm, WV (Water Vapor) 6500 - 7000 nm, TIR-1 (Thermal Infrared) 1020 - 1120 nm, TIR-2 (Thermal Infrared) 1150 - 1250 nm) Mahammad; Kaushik. The Visible and SWIR bands are not available at nighttime observations, and the images in all infrared bands are visually similar, thus in this work, the thermal infrared (TIR1) band has been used. In the following sub-sections, all the relevant information regarding the INCYDE dataset is presented.

## 3.1 DATASET CURATION PIPELINE

In order to create an end-to-end cyclone intensity estimation solution, the task of developing the solution can be divided into two parts, i.e., object detection and intensity estimation. In the first step, an object detection algorithm can be used to identify and localize cyclones in the image, and in the second step, the cropped cyclone image can be used as input to the intensity estimation model to get the maximum sustained wind speed (MSWS) of the cyclone. Thus, the entire solution is able to output cyclone intensity as well as its location in the image. This pipeline also allows focusing research on cyclone detection and intensity estimation independently. However, other design paradigms can also be explored, the dataset includes a CSV file with cyclone bounding box information as well as cyclone intensity that may help researchers experiment with different design choices for the solution pipeline. In this work, we have presented a dataset with both types of annotations. Figure 1 shows the process we used to curate the dataset. We have used INSAT 3D/3DR Mahammad; Kaushik satellite infrared images for cyclone periods from 2013-2021, and for the labels, we have used IMD best track data IMD (2022) that contains relevant information regarding cyclones from 1982 to 2022. The data from INSAT 3D/3DR has been transformed from GeoTIFF format to JPG for the purpose of training baseline models and to help train deep learning models in a reasonable time, the dataset however is also available in GeoTIFF format. The best track data is first compiled for each year, cleaned, and then standardized in a CSV format. Additional cyclone label data has been generated using interpolation of the existing best track data. The combined data is then split into training, validation and testing sets according to the distribution of cyclone categories in the dataset. The training set is also augmented using various augmentation techniques. The process of interpolation, dataset augmentations, and dataset splits are explained thoroughly in later sections. Similarly, the cyclone detection annotations in COCO format are made using the best track data, more details are discussed in later sections.

## 3.2 IMAGE CROPPING AND BOUNDING BOX DETAILS

In this work, we propose a novel dataset called INCYDE, which stands for INSAT-based Cyclone Detection and Intensity Estimation. Figure 2 shows some sample images from the proposed INCYDE dataset for cyclone detection from satellite IR imagery. Similarly, Figure 3 shows some sample images from the proposed INCYDE dataset for cyclone intensity estimation, the cropped images are taken from full INSAT 3D/3DR imagery and cropping 15 degrees latitude and longitude around the cyclone center in the satellite image. The distance between 2 nearby pixels in a satellite image corresponds to approximately 0.034 degrees in latitude for height and about 0.038 degrees in longitude for width. To select an area of interest, we choose a range of 7.5 degrees in latitude both above and below the center, which corresponds to approximately $(7.5/0.034) \times 2 \approx 437$ pixels in height. Similarly, we choose a range of 7.5 degrees in longitude on either side of the center, corresponding to approximately $(7.5/0.038) \times 2 \approx 398$ pixels in width. The INCYDE dataset presents the intensity estimation annotations in CSV format and cyclone detection annotations in COCO Lin et al. (2014) format. The bounding boxes are generated using the center of the cyclone as a reference

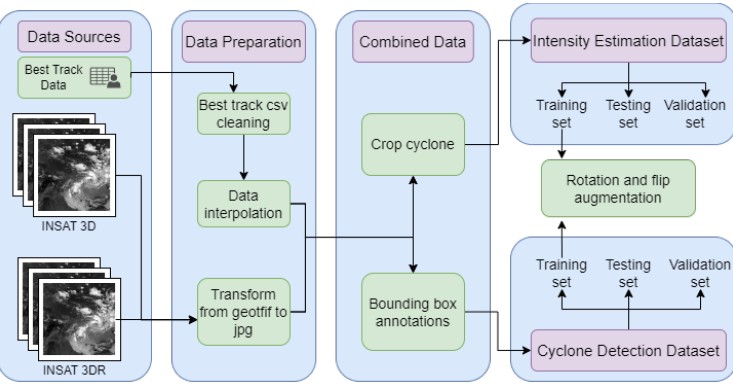

Figure 1: Workflow for curating the INCYDE dataset from best track data and satellite images for cyclone detection and intensity estimation task

point. The center of the cyclone is taken from IMD best track data. A bounding box is then created with dimensions of approximately 7.5 degrees apart from the center, which is similar to the cyclone crop dimensions. This translates to approximately 437 and 398 pixel values for the height and width respectively of the bounding box. The coordinates of the bounding box, including its lower left corner, height, and width, are exported in COCO format to a JSON file for the object detection task.

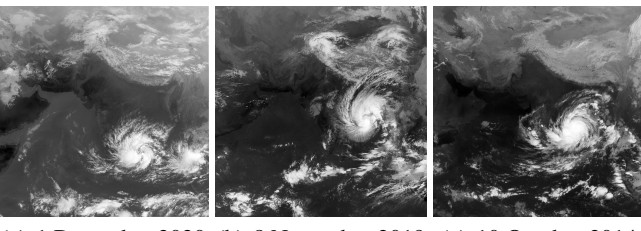

(a) 1 December 2020  (b) 8 November 2019  (c) 10 October 2014

Figure 2: Full infrared satellite images during cyclone period without crop

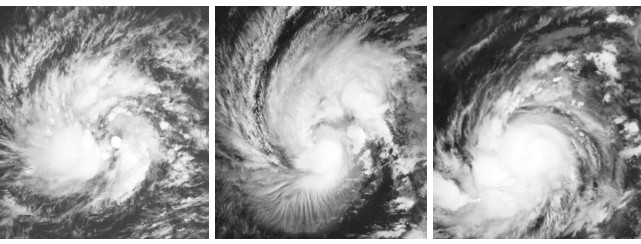

(a) 1 December 2020  (b) 8 November 2019  (c) 10 October 2014

Figure 3: Cropped cyclone images

### 3.3 Interpolation of Labels in the INCYDE dataset

The INSAT 3D/3DR satellite images are in the interval of 30/15 minutes but the IMD best track data is the interval of 6 hours. So, in order to efficiently use all the infrared satellite imagery available for a particular cyclone, it is imperative to use satellite images with timestamps lying in between consecutive best-track data labels. We used interpolation to generate more labels from existing labels for the satellite images. Figure 4 shows the process of interpolation of labels. First, the stepsize is calculated using two consecutive rows in the best track data for latitude, longitude, and MSWS fields. The stepsize is then added to the original row of best track data in multiples of the number of steps in between two consecutive timestamps. The process is then continued for the entire best track data to generate over 20k labels from existing 4k labels.

### 3.4 Dataset Augmentation

Augmentation has been used to increase the size of the dataset. We have used openCV Bradski (2000) and PIL Clark (2015) libraries to augment the images. Figure 5 shows augmented satellite images for cyclone detection in the proposed INCYDE dataset. We have performed $90°$ rotation, $180°$ rotation and $270°$ rotation. Using these augmentations the resulting dataset size increased from 21k images to over 68k images. Similarly, the dataset was also augmented for the cyclone intensity estimation task using the same augmentations.

### 3.5 Dataset Split

The dataset has been split into training, validation, and testing sets based on sequences of cyclones in the dataset. Every storm was given a unique ID and the highest grade it reached. According to the highest wind speed recorded, the cyclone sequences were assigned classes based on the IMD best

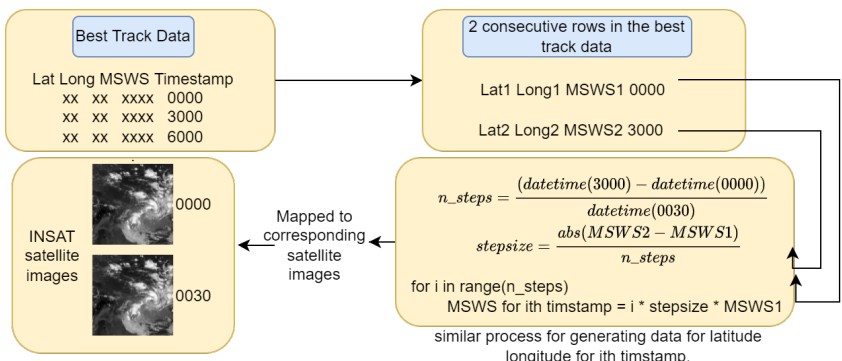

Figure 4: Process of generating new labels through interpolation of existing labels

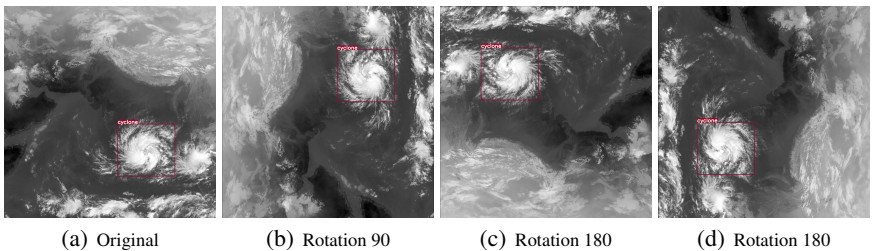

(a) Original      (b) Rotation 90      (c) Rotation 180      (d) Rotation 180

Figure 5: Augmented infrared satellites images for the INCYDE dataset with the bounding box for the cyclone in the image

track cyclone classification IMD (2022). The dataset was divided to ensure equal representation of the different classes of cyclone storms, for instance, for the highest class of cyclone i.e., Super Cyclonic Storm (SuCS), there were only 4 cyclone storm sequences in the combined dataset, the split was made in such a way that the final training set contains 2 SuCS sequences, validation, and testing sets get 1 SuCS sequence each. The rest of the dataset was split into the train, test, and validation sets by randomly sampling 70% of the sequences into the train, 15 % into the test, and 15% into the validation set grouped by different classes. To avoid data leakage in the dataset, it was crucial to split the data in a way that ensures an image from one sequence is not included in more than one set, given the fact that consecutive images in a sequence are highly similar. The split was performed in this way so as to create completely unseen sets for test and validation set as consecutive images in a sequence of a cyclone have very similar images and thus randomly sampling all the images in a naive approach would result in data leakage. The final split resulted in 16k training set images, 2.5k test set images, and 2.39k validation set images before augmenting the training dataset.

## 4    DATASET STATISTICS

This section presents some statistics about the proposed INCYDE dataset. Figure 6 shows the histogram of mean sustained wind speed (MSWS) which is a metric for cyclone intensity in nautical miles per hour (knots) for the proposed INCYDE dataset and other datasets for cyclone intensity estimation as found in the literature. It can be observed in the figure that a lot of images have cyclones with speeds around 20-40 knots and very less images have cyclone intensity higher than 100 knots and it is consistent across all datasets for cyclone intensity estimation. It is in line with the fact that in the entire lifecycle of a cyclone, for most of the part, the cyclone has speeds in the range of 20 to 40 knots, and for very little time the cyclone actually achieves the highest MSWS at its peak. Detection and intensity estimation of cyclones during the first phase is crucial for building a robust early warning system for cyclone prediction as during the initial phases, the cyclone has not yet formed spiral-like cloud patterns that are associated with cyclonic storms.

Table 1 shows the size of the proposed INCYDE dataset in comparison to other (SOTA) cyclone intensity estimation datasets. It can be observed that our proposed INCYDE dataset has a higher number of images in the training dataset and the overall number of images are comparable to Maskey et al. In this table, we have also shown the number of images for the cyclone detection dataset. There is a slight difference between the number of images in cyclone detection and intensity estimation datasets as in some satellite images multiple cyclones have appeared at a single timestamp that resulted in 1 image for cyclone detection corresponding to multiple cropped images for cyclone intensity estimation. It can also be observed that our proposed dataset has a higher image size as we did not downsample the original image to preserve the finer details that would help develop better models for cyclone detection and intensity estimation. All of the cyclone intensity estimation datasets use infrared (IR) satellite images. Our dataset surpasses prior works by expanding the geographical coverage of existing datasets in this field and providing high-resolution satellite images without downsampling. It also includes a cyclone detection dataset in COCO format, making it a valuable resource for researchers in the field.

## 5 Experiments

Along with the dataset, several baseline models have been trained on the proposed dataset for both cyclone detection and cyclone intensity estimation task. For object detection, YOLOv5Jocher & et al. (2021), EfficientDetTan et al. (2020), Faster RCNNRen et al. (2015) have been trained as baselines. YOLOv5Jocher & et al. (2021) is an efficient single-stage anchor-based object detector that uses a feature pyramid network PANet as the backbone. Faster-RCNN Ren et al. (2015) is a two-stage object detector that uses a region proposal network in first stage to find the region of interest, and in the second stage, it performs object detection in order to be more accurate. EfficientDet Tan et al. (2020) is another one-stage anchor-based object detector developed on top of efficient net Tan & Le (2019) backbone. For cyclone intensity estimation, ResNet He et al. (2015), Inception-V3 Szegedy et al. (2015b), EfficientNet Tan & Le (2019), DenseNet-121 Huang et al. (2017), MobileNet-V2 Sandler et al. (2018) with modified prediction heads to output a single value for intensity estimation are trained on the INCYDE dataset.

Table 1: Cyclone intensity estimation dataset sizes and statistics

| Dataset | Train | Test | Validation | Image Size | Total Frames | IR Band |
|---|---|---|---|---|---|---|
| Maskey et al. (2020) | 70258 | 44378 | – | 366 x 366 | 114636 | IR |
| TCIR Chen et al. (2018) | 70501 | – | – | 201 x 201 | 70501 | IR/PMW |
| INCYDE (Intensity Estimation) | 65644 | 2503 | 2399 | 398 x 437 | 68912 | TIR |
| INCYDE (Cyclone Detection) | 64152 | 2503 | 2399 | 1618 x 1616 | 67420 | TIR |

### 5.1 Evaluation metrics

In this work, the proposed dataset uses root mean square error (RMSE) as the evaluation metric for the cyclone intensity estimation task and mean average precision (mAP) as the evaluation metric for

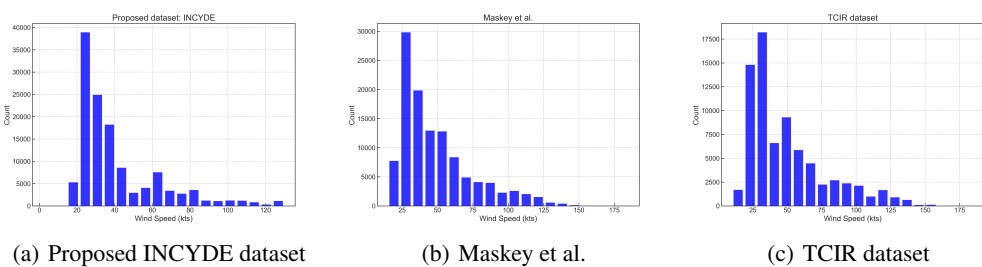

(a) Proposed INCYDE dataset  (b) Maskey et al.  (c) TCIR dataset

Figure 6: Histogram of mean sustained wind speeds in knots for various cyclone intensity estimation datasets in literature

the cyclone detection task. RMSE as the name implies is calculated by taking the square root of the mean of the squared differences between the predicted values and the actual observed values. RMSE is calculated using the following formula:

$$RMSE = \sqrt{\Sigma(pr - ob)^2}$$

where, pr = predicted value and, ob = observed value

Mean average precision (mAP) is a common metric used to evaluate the performance of object detection models. The mAP is calculated by first determining the precision and recall for each class in the dataset. Precision measures the accuracy of the model in identifying true positives, while recall measures the completeness of the model in identifying all positives. For object detection, the true positive is calculated using Intersection over Union (IoU) of bounding boxes of predicted and ground truth bounding boxes. The precision and recall values are then used to calculate the Average Precision (AP) for each class. AP is the area under the precision-recall curve for each class. The mAP is then calculated as the average of the AP values across all classes. This provides an overall measure of the model's performance.

## 5.2 RESULTS AND DISCUSSIONS

For baseline models, a few algorithms were trained on the proposed dataset for cyclone detection as well as intensity estimation tasks. The trained models with their respective accuracy metrics are shown below. The models were trained on a subset of the entire INCYDE, specifically, these models were trained only on the dataset without augmented images as the size of the augmented dataset makes it impossible to train the baseline models in reasonable time with limited resources, so we have kept the task of training the baseline models on an augmented dataset for future work. For cyclone detection, i.e., an object detection task, the mean average precision (mAP) metric is used while for intensity estimation, i.e., a regression task, root mean squared error (RMSE) is used to report the results.

Table 2 shows the mean average precision (mAP) of YOLOv5, Faster-RCNN, and EfficientDet on the proposed dataset along with a YOLOv5 model trained on the augmented INCYDE dataset. It can be observed that the mAP of SOTA object detectors hovers around 40-63 mAP on the validation set which acts as a solid baseline for future work. In our study, we found out that the models trained on the dataset without augmentation performed well on non-augmented images but performed poorly on augmented images, while the YOLOv5 trained on augmented INCYDE dataset dropped in terms of mAP as compared to YOLOv5 trained on non-augmented INCYDE dataset but the YOLOv5-Aug, when used to inference on the augmented dataset, performed much better implying the use of augmentation in the proposed INCYDE dataset actually helped train a better generalizable model. Figure 7 shows the inference of YOLOv5 trained on the INCYDE dataset with augmentation, it can be observed that the model is able to detect cyclones anywhere in the image, implying better generalization ability.

Table 3 shows the RMSE of ResNet-18, Inception-v3, EfficientNet, DenseNet-121, and MobileNet V2 trained on the proposed dataset. In table 3, it can be observed that the best RMSE achieved was with MobileNet-V2 at 15.44 knots RMSE which acts as a solid baseline for future work. The RMSE of MobileNet-v2 on the proposed INCYDE dataset is comparable to baseline models on other related datasets that hover around 13 knots. Maskey et al. (2020) were able to achieve an RMSE of

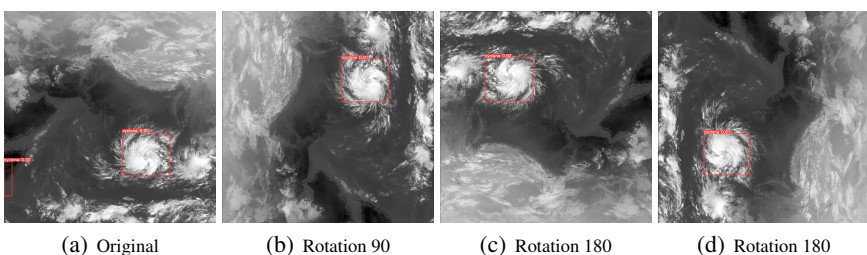

(a) Original        (b) Rotation 90        (c) Rotation 180        (d) Rotation 180

Figure 7: YOLOv5 trained on augmented INCYDE dataset inferred on satellite infrared images

13.24 knots, and Chen et al. (2018) were able to achieve an RMSE of 10.6 knots. However, direct comparison of baseline models trained on different datasets is not appropriate, but it does provide a benchmark to compare against.

Table 2: Summary of baseline models for object detection

| Model Name | Validation mAP | Epochs |
|---|---|---|
| YOLOv5 Jocher & et al. (2021) | 63.35 | 20 |
| Faster Ren et al. (2015) RCNN | 48.5 | 20 |
| EfficientDet Tan et al. (2020) | 40.5 | 20 |
| YOLOv5-Aug Jocher & et al. (2021) | 42.37 | 20 |

Table 3: Summary of baseline models for intensity estimation

| Model Name | RMSE Test (knots) | Epoch |
|---|---|---|
| Resnet-18 He et al. (2015) | 18.78 | 60 |
| Inception-v3 Szegedy et al. (2015a;b) | 18.28 | 60 |
| EfficientNet Tan & Le (2019) | 15.28 | 13 |
| DenseNet-121 Huang et al. (2017) | 15.55 | 60 |
| MobileNet V2 Sandler et al. (2018) | 15.44 | 60 |

## 6 CONCLUSION AND FUTURE DIRECTIONS

In this work, a novel dataset has been presented for cyclone intensity estimation and cyclone detection. The dataset contains INSAT 3D/3DR thermal infrared satellite images since the year 2013 to 2021. The proposed INCYDE dataset is curated using a series of steps involving data collection, data cleaning, label interpolation, data augmentation, and data split. A few SOTA object detection algorithms are trained on the proposed INCYDE dataset to act as baselines. A few modified deep CNN models are also trained on the proposed INCYDE dataset for cyclone intensity estimation to act as baselines for cyclone intensity estimation. The INCYDE dataset contains over 21k images of cyclones with their annotations (68k with augmentations) in bounding box configuration for object detection in COCO format as well as for single-valued intensity estimation tasks in a CSV format. The dataset is comparable to other cyclone intensity estimation datasets in the literature. The IN-CYDE dataset would help researchers develop SOTA models for cyclone detection and intensity estimation using innovative techniques which in turn would be used as an early warning system for cyclones.

In the future, there lies potential in investigating the influence of diverse augmentation techniques on the efficiency of cyclone intensity estimation models. Additionally, our dataset's inclusion of temporal information about cyclones opens avenues for delving into the realm of time series modeling. The temporal information also opens up avenues for cyclone track prediction using satellite imagery. This direction holds promise for deeper insights into cyclone behavior over time. Furthermore, given the dataset's dual offering of both cyclone detection and intensity estimation datasets, there exists an opportunity to explore a unified approach. This entails venturing into a one-stage integrated solution for cyclone detection and intensity estimation. Such an exploration could pave the way for a more comprehensive analysis. Since the dataset pertains to one geographical region, it would be interesting to investigate the performance of a model on a geographical region that is different than the one the model was trained on. These potential avenues for future research highlight the dynamic nature of our dataset's applications and its capacity to drive innovative advancements in cyclone analysis.

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
