# OpenReview forum: "INCYDE: A large scale cyclone detection and intensity estimation dataset using satellite infrared imagery"
_ICLR.cc/2024/Conference — ICLR 2024 Conference Desk Rejected Submission_

### Official Review · Reviewer_Vk2U · 2023-10-23

**Soundness:** 2 fair
**Presentation:** 3 good
**Contribution:** 2 fair
**Rating:** 3
**Confidence:** 4

**Summary:**

The main contribution of the paper is a dataset of cyclone tracks over the Indian Ocean region identified from INSAT 3D/3DR satellite. The tracks have been identified after a significant amount of preprocessing, including temporal interpolation of the satellite image sequences. Based on this dataset, the authors have posed two problems: localization of the cyclone's position, and estimation of its intensity. Based on intensity, each even has been categorized as deep depression, sever cyclonic storm etc. They can be solved using Deep Learning models, similar to the object detection techniques like YOLO, RCNN etc which the authors have tested as benchmarks on this dataset.

**Strengths:**

The paper's main contribution is a dataset of cyclone tracks over the Indian Ocean region. such datasets, especially coupled with high-quality ground-truth labels as the authors have provided, are very useful to meteorology research.

**Weaknesses:**

The main weakness is that, there are no technical contributions related to ML
The authors have provided benchmarks using models like RCNN and YOLO. However, these are standard general-purpose object detection models. There has been significant research in recent years, using more advanced DL models for cyclone track prediction. Admittedly, those track predictions are not necessarily based on IR satellite imagery which the authors have done. However, to provide the context, the authors should have discussed those regarding the benchmarking. Furthermore, computer vision also has significant research on object tracking in videos, which is more suitable to this application than image-by-image detection as done by YOLO or RCNN. The authors have not used those for benchmarking, making the paper rather weak for ICLR.

For example, some important references can be:
1) Deepti: Deep-Learning-Based Tropical Cyclone Intensity Estimation System by Maskey et al, IEEE Journal of Selected Topics in Applied Earth Observations and Remote Sensing, 2020.
2) A Center Location Algorithm for Tropical Cyclone in Satellite Infrared Image by Wang et al, EEE Journal of Selected Topics in Applied Earth Observations and Remote Sensing, 2020.
3) MGTCF: Multi-Generator Tropical Cyclone Forecasting with Heterogeneous Meteorological Data by Huang et al, AAAI 2023

**Questions:**

No questions as such.
My general comment is that the dataset is definitely a very useful contribution for meteorology/climate studies, and the work is very suitable for conferences on Earth Observation Systems, or Climate Informatics.
But at least in the current form, I do not think it will be of interest to a sufficient section of the ICLR audience. Cyclones are very complex structures, and developing new representations for them through Neural features from IR Satellite imagery for detection, tracking and intensity estimation would have made a solid contribution. I encourage the authors to work on this.

---

> ### Author Response · Authors · 2023-11-22
> **Response to the comments of reviewer Vk2U**
>
> > ‘The main weakness is that, there are no technical contributions related to ML The authors have provided benchmarks using models like RCNN and YOLO. However, these are standard general-purpose object detection models. There has been significant research in recent years, using more advanced DL models for cyclone track prediction. Admittedly, those track predictions are not necessarily based on IR satellite imagery which the authors have done. However, to provide the context, the authors should have discussed those regarding the benchmarking. Furthermore, computer vision also has significant research on object tracking in videos, which is more suitable to this application than image-by-image detection as done by YOLO or RCNN. The authors have not used those for benchmarking, making the paper rather weak for ICLR.
> For example, some important references can be:
> >1. Deepti: Deep-Learning-Based Tropical Cyclone Intensity Estimation System by Maskey et al, IEEE Journal of Selected Topics in Applied Earth Observations and Remote Sensing, 2020.
> >2. A Center Location Algorithm for Tropical Cyclone in Satellite Infrared Image by Wang et al, EEE Journal of Selected Topics in Applied Earth Observations and Remote Sensing, 2020.
> >3. MGTCF: Multi-Generator Tropical Cyclone Forecasting with Heterogeneous Meteorological Data by Huang et al, AAAI 2023’
>
> Thank you for your thoughtful comments. While we acknowledge the importance of cyclone tracking, our paper focuses on cyclone intensity estimation and provides an additional dataset for cyclone detection to facilitate the development of end-to-end models. We have included temporal information in the dataset in the form of filenames, allowing researchers to construct their tracking datasets if they choose to do so. As you rightly mentioned, track prediction can be conducted independently of satellite images, thus we have developed a dataset primarily focusing on cyclone detection and intensity estimation and not on cyclone track prediction. Regarding the suggestion to use object tracking in videos as benchmarking for our dataset, we believe the two tasks are considerably different. Significant research has taken place in improving the inference time of object tracking algorithms in videos, but the dataset and satellite imagery we utilize have a frequency of 15 minutes per image. Additionally, research in object tracking in videos involves accurately tracking multiple objects through occlusion at once, but the object in our work (cyclone) occurrences are infrequent, with at most two cyclones in an image. Thus, the challenges addressed in general object-tracking research, such as tracking multiple objects, fast inference, occlusion, etc. in an image, are not prevalent in our domain. However, we encourage researchers interested in cyclone tracking to create their datasets tailored to their specific needs and we have now mentioned this in the conclusion and future directions section (section 6) as well. Also, thanks for bringing the valuable references to our attention. Regarding the references that you have provided, we have already cited the first paper in your list in our paper. While the second paper is relevant, the authors have not provided a publicly accessible link to their dataset. The third paper, although very recent, does not pertain to cyclone intensity estimation or cyclone detection. We appreciate your feedback and remain open to further discussions or suggestions.

---

> > ### Author Response · Authors · 2023-11-22
> > **Response to the comments of reviewer Vk2U part 2**
> >
> > >‘No questions as such. My general comment is that the dataset is definitely a very useful contribution for meteorology/climate studies, and the work is very suitable for conferences on Earth Observation Systems, or Climate Informatics. But at least in the current form, I do not think it will be of interest to a sufficient section of the ICLR audience. Cyclones are very complex structures, and developing new representations for them through Neural features from IR Satellite imagery for detection, tracking and intensity estimation would have made a solid contribution. I encourage the authors to work on this.’
> >
> > We appreciate your feedback. While we acknowledge that this work is also suitable for earth observation or remote sensing conference/journals, we intentionally submitted our work at a Machine Learning conference to bring a remote sensing problem to the computer vision/ML research community and we have also mentioned this in section 1 introduction. The curated dataset for cyclone detection and intensity estimation, along with temporal information, offers an opportunity for the computer vision research community to contribute valuable insights, we encourage and welcome the research community to explore various models and approaches for cyclone detection, tracking, and intensity estimation using our dataset. We fully agree with your suggestion regarding the exploration of new representations for cyclones through Neural features from IR Satellite imagery. We plan to work on this in the future using our dataset.

---

> > > ### Comment · Reviewer_Vk2U · 2023-11-22
> > >
> > > I appreciate the reasons for submitting to an ML venue. I am myself a ML/computer vision-turned-climate researcher, and I understand fully that it is a worthy effort to release a dataset like this for ML/vision researchers to take forward. Unfortunately, the paper does not seem a good fit for this specific venue, where the focus on representations of complex data for ML.

---

> > ### Comment · Reviewer_Vk2U · 2023-11-22
> >
> > Thank you for your response. However, I do not agree with your comments on tracking. 15 minutes per image seems to be plenty of time to do tracking. Tracking does not need multiple targets, though it can work in presence of multiple targets. Occlusion is a challenge, not a necessity. In your scenario, tracking is actually easier.

---

> > > ### Author Response · Authors · 2023-11-23
> > >
> > > Thank you for your feedback. While we have not incorporated cyclone tracking in this work, we acknowledge the potential for future work in this direction. Researchers can already leverage our dataset’s temporal information to explore and implement tracking techniques. We appreciate your guidance on creating a benchmark for tracking and we will definitely incorporate this in future.

---

### Official Review · Reviewer_TKNa · 2023-10-28

**Soundness:** 1 poor
**Presentation:** 1 poor
**Contribution:** 1 poor
**Rating:** 1
**Confidence:** 5

**Summary:**

The paper proposes a dataset for cyclone detection and evaluates several standard object detection and regression methods for this task.
It establishes that the Dvorak technique is usually used in the field. However, it is not compared to the experiments. Hence, it is not clear if any of the tested models is applicable or an improvement in this field. I believe this would be a good/reasonable paper for a remote sensing or application-specific conference or journal. However, the value for ICLR is not clear.

There are also several points in the text that hint towards a lack of care in this submission.
* The introduction makes several high-level statements without citations. E.g., "In literature, there have been some well-documented datasets for cyclone intensity estimation". However, none are cited.
* citations are stated twice, such as "Maskey et al. Maskey et al. (2023)" throughout the entire paper.
* the paper is a half-page over length.

This lack of quality is surprising given that this paper is a resubmission and was previously rejected. However, to my knowledge, little or no feedback or input from previous reviews has been implemented. It appears to me that the authors resubmitted the identical text without re-reading the compiled PDF, checking the citations, or modifying the length.

**Strengths:**

* important problem, but not a topic for this conference.

**Weaknesses:**

* lack of care in the submission
* no methodological insights
* no comparison to the standard Dvorak method

**Questions:**

* What are exactly the changes made between the previously rejected version and this one? How were the reviewer comments from that previous version considered in this resubmission?
* Several high-profile domain/application journals exist where the challenges of cyclone detection can discussed with reviewers that have domain expertise in exactly this problem field. For instance, the IEEE Journal of Selected Topics in Applied Earth Observations and Remote Sensing, where Manil Maskey (cited extensively in this work) submitted their paper. What makes ICLR the suitable venue for this application/problem? Additionally, not a single paper cited in this work was published at ICLR.

---

> ### Author Response · Authors · 2023-11-22
> **Response to the comments of reviewer TKNa**
>
> > ‘The paper proposes a dataset for cyclone detection and evaluates several standard object detection and regression methods for this task. It establishes that the Dvorak technique is usually used in the field. However, it is not compared to the experiments. Hence, it is not clear if any of the tested models is applicable or an improvement in this field. I believe this would be a good/reasonable paper for a remote sensing or application-specific conference or journal’
>
> Thank you for your feedback on our paper. In our work, the labels for the dataset were originally generated using the Dvorak technique by the Indian Meteorological Department (IMD), more details can be found here https://rsmcnewdelhi.imd.gov.in/report.php?internal_menu=MzM= and https://mausam.imd.gov.in/imd_latest/contents/pdf/cyclone_sop.pdf. The manual nature of the Dvorak technique introduces subjectivity and limitations, prompting the adoption of AI methods to automate and provide more objective insights into cyclone detection and intensity estimation. We agree that a remote sensing or application-specific venue would also be appropriate for this paper but through this work, we aim to bring a remote sensing problem to the computer vision/ML research community and we have also mentioned this in our paper in the section 1 introduction. We have curated a dataset for cyclone detection and cyclone intensity estimation and we believe that the ML research community has a lot to offer if they are provided with a comprehensive dataset for cyclone detection and intensity estimation.
>
> >‘The introduction makes several high-level statements without citations. E.g., "In literature, there have been some well-documented datasets for cyclone intensity estimation". However, none are cited.’
>
> Thank you for your feedback on our paper. We want to clarify that we have indeed cited publicly accessible datasets in our paper, including TCIR and Maskey et al., we have even compared our dataset with other publicly accessible datasets to provide an objective comparison between these datasets as shown in table 1 on page 7.
>
> >‘citations are stated twice, such as "Maskey et al. Maskey et al. (2023)" throughout the entire paper.'
>
> >'the paper is a half-page over length.'
>
> Thank you for bringing the issue of improper citations to our attention. We sincerely apologize for the oversight. The repetition of citations was inadvertently introduced due to the specific citation format required by ICLR which was considerably different than the previous conference. We have now rectified this issue, ensuring that citations appear correctly throughout the entire paper. We have also taken the necessary steps to trim down the content, and the revised version now adheres to the specified 9-page limit.
>
> >‘Several high-profile domain/application journals exist where the challenges of cyclone detection can discussed with reviewers that have domain expertise in exactly this problem field. For instance, the IEEE Journal of Selected Topics in Applied Earth Observations and Remote Sensing, where Manil Maskey (cited extensively in this work) submitted their paper. What makes ICLR the suitable venue for this application/problem? Additionally, not a single paper cited in this work was published at ICLR.’
>
> We appreciate your suggestion and acknowledge that there are domain-specific journals for discussing challenges in cyclone detection, such as the IEEE Journal of Selected Topics in Applied Earth Observations and Remote Sensing. However, our choice of ICLR stems from our objective of bridging the gap between remote sensing and the computer vision/ML community, we have also mentioned the purpose of this paper being bridging the gap between remote sensing and AI research in our paper in section 1 introduction. By presenting our work at a machine learning conference, we aim to engage the computer vision research community in addressing the unique challenges of cyclone detection and intensity estimation.

---

> > ### Author Response · Authors · 2023-11-22
> > **Response to the comments of reviewer TKNa part 2**
> >
> > >‘This lack of quality is surprising given that this paper is a resubmission and was previously rejected. However, to my knowledge, little or no feedback or input from previous reviews has been implemented. It appears to me that the authors resubmitted the identical text without re-reading the compiled PDF, checking the citations, or modifying the length.’
> >
> > >‘What are exactly the changes made between the previously rejected version and this one? How were the reviewer comments from that previous version considered in this resubmission?’
> >
> > This work was earlier submitted to another conference where we received comprehensive feedback from the reviewers and since then we have worked on the comments to improve our paper. We have incorporated many changes from the feedback of previous reviewers. We acknowledge that there was an oversight on our part in proofreading the compiled PDF, we have now fixed the citations and we have also trimmed the paper to fit into the 9-page limit. It might not be possible to provide all the comments and responses of previous reviewers, but we can provide a brief overview of the most prominent changes incorporated from previous reviewers comments:
> > - Earlier, we had flip and rotation augmentations in our dataset, but the comments suggested the flip augmentations would not be appropriate for cyclones, so we have removed the flip augmentations, and we have also updated the numbers everywhere to reflect the correct size of the dataset.
> > - One reviewer suggested adding future directions which were missing earlier in the section titled ‘Conclusion and future directions’.
> > - One reviewer asked us to provide more context about the Dvorak technique, so we have added the necessary details.
> > - The reviewers felt that the details about cropping the cyclone image and creating bounding boxes were inadequate, so we added a dedicated section about the process of cropping the cyclone from the image and creating bounding boxes for the cyclone detection task.
> > - Added some details about baseline models in other datasets
> > - Some minor changes, grammar corrections, and some modifications in the overall organization of the paper
> >
> >  These are a few of the many changes that were made during the rebuttal phase of the previous conference where this work was submitted earlier.

---

> > > ### Comment · Reviewer_TKNa · 2023-11-22
> > > **Changes with respect to earlier submission**
> > >
> > > Thank you for providing an overview of changes to this version and the previous submission.
> > >
> > > I see that some general changes have been made between the submissions. I apologize that I may have been too conclusive in parts in my first review in this regard. Still, the text-wise presentation made me think the text had been copied between submission templates. Concretely, these were word-by-word identical abstracts, and errors in the citation style and the over-length submission in this submission that hinted towards a mere copying of the text between the submission templates.
> > > Given the many constructive comments and suggestions in the previous version (similar to the comments by other reviewers in this submission), would have required a more thorough rewriting of the entire paper including the abstract.
> > >
> > > Also here, some of the listed changes, such as the explanation of the context of the Dvorak technique, do not seem to be addressed sufficiently, as this question was raised again in this submission.
> > > Here, the response to my raised question with external links and a rather motivational description like "It introduces subjectivity and limitations" seems not satisfactory in this regard. From this response, it seems that the Dvorak technique is rather a manual labeling tool than a detection approach ("manual nature", "introduces subjectivity", "labels generated through the Dvorak technique")? The paper abstract suggests that it is rather a [automated] detection method.
> > > This should become clear from the description within the abstract and paper without referencing external sources.

---

> > > > ### Author Response · Authors · 2023-11-23
> > > >
> > > > Thank you for your constructive feedback. We have updated the abstract to better reflect the manual nature of the Dvorak technique. You are right that the Dvorak technique is a manual labeling technique, the analysts follow a series of rules to estimate cyclone intensity by looking for certain cloud features observed in satellite images. We also want to highlight that we have already addressed the manual nature of the Dvorak technique in paragraph 2 of the related works section (section 2). Also, we acknowledge the importance of clarity in the presentation of our paper, and thus since the previous conference, we have reorganized our paper including introducing new sections for a more cohesive structure. We welcome any further recommendations or suggestions for improvements in the organization of the paper.

---

> > ### Comment · Reviewer_TKNa · 2023-11-22
> > **Thank you for the responses**
> >
> > Thank you for responding to the review.
> >
> > I fully agree and highly encourage trans-disciplinary exchange across research fields. In particular between remote sensing and machine learning.
> >
> > However, I also believe such trans-disciplinary contributions must be useful and applicable for both research fields.
> > After reading the paper again, I don't see an immediate and concrete benefit of what method-focused problems machine learning researchers could investigate on this dataset aside from testing ML algorithms on an environmentally important application.
> >
> > I believe organizing or submitting to trans-disciplinary workshops or organizing challenges would be a better vehicle than submitting such a problem-specific dataset paper at a top-tier ML main conference.

---

### Official Review · Reviewer_VDYR · 2023-11-02

**Soundness:** 2 fair
**Presentation:** 2 fair
**Contribution:** 3 good
**Rating:** 5
**Confidence:** 4

**Summary:**

The paper presents a novel infrared cyclone dataset for cyclone prediction. The dataset covers Indian Ocean region. It includes raw (21K) and augmented (up to 68K) infrared satellite imagery data and annotations for 2013 - 2021 years. The authors include the features, required for object identification and intensity estimation and cyclone track data.

The main contribution of the paper is the dataset itself over the new geographical region; the dataset creation pipeline and a comparison of state-of-the-art models, presented in this dataset. The dataset creation pipeline is well presented, and the paper is generally well written. The methods, presented in the paper, cover three popular deep learning models for object detection (Yolo5, EfficientNet and Faster RCNN) and five popular deep learning methods for intensity estimation.

The overall paper presentation and problem description, as well as the dataset itself, meet the minimum criteria for the novel contribution in the field of ML for environmental tasks.

**Strengths:**

The main strength of the paper is in the presentation of the novel dataset, which is extremely useful for research scientists.

**Weaknesses:**

The two main weaknesses are (1) the absence of the link to the dataset itself (or at least I was not able to easily identify it in the text) and (2) overlap with some recently published work on the similar topic which is not cited in this submission.

**Questions:**

1.	Please present the dataset for the download and review as the paper describes the dataset.
2.	There are some typos and a problem with the citations, which makes is difficult to read the text.
3.	Please present the overview of the relevant literature (e.g. same satellite + same dataset + same methods). I understand that the exact topic of the paper is different, but the novelty of this current work should be stated explicitly.
4.	The paper exceeds the 9-page limit.

**Details Of Ethics Concerns:**

There are a few published papers with the same last author, which were journal publications recently and closely resemble parts of this work. Also, I could not find the link to the dataset itself, so I don't understand whether the dataset is the same or not

https://ijrpr.com/uploads/V4ISSUE4/IJRPR11379.pdf

https://ijrpr.com/uploads/V4ISSUE4/IJRPR12056.pdf

---

> ### Author Response · Authors · 2023-11-21
> **Response to the comments of reviewer VDYR**
>
> > ‘Please present the dataset for the download and review as the paper describes the dataset.’
>
> We appreciate your feedback regarding access to the dataset and we apologize for not including it in our original manuscript. Due to a lack of clarity with the double-blind policy, we did not originally include the dataset link in our submission. We have updated the manuscript with an anonymized link (Zenodo: 10.5281/zenodo.8015544) to enable verification of our methodology.
>
> > ‘There are some typos and a problem with the citations, which makes it difficult to read the text.’
>
> Thanks for bringing this to our notice, we realized there were some mistakes with how the citations were presented and we apologize for the typos and the problems with the citations, we have now fixed the typos and rectified the problem with citations in the manuscript.
>
> > ‘The paper exceeds the 9-page limit.’
>
> We have trimmed some part of the paper that we felt was not relevant, specifically, we have removed the figure about the temporal distribution of a few cyclones in the dataset (formerly figure 6), and we have removed cyclone classification based on wind speed which we felt would be not relevant to computer vision/ ML research community (formerly figure 8).
>
> >‘Please present the overview of the relevant literature (e.g. same satellite + same dataset + same methods). I understand that the exact topic of the paper is different, but the novelty of this current work should be stated explicitly.’
>
> >‘overlap with some recently published work on the similar topic which is not cited in this submission.’
>
> >‘There are a few published papers with the same last author, which were journal publications recently and closely resemble parts of this work. Also, I could not find the link to the dataset itself, so I don't understand whether the dataset is the same or not
> https://ijrpr.com/uploads/V4ISSUE4/IJRPR11379.pdf
> https://ijrpr.com/uploads/V4ISSUE4/IJRPR12056.pdf’
>
> We haven't encountered any literature with the same satellite + same dataset using the same methods, we have gone through the references that you have provided, and we believe the references contain unsupported assertions and inaccurate information, and no reproducibility in terms of dataset and ML models. For instance, the authors of the paper https://ijrpr.com/uploads/V4ISSUE4/IJRPR11379.pdf have not provided any publicly accessible link to the dataset nor have they provided any details about how the dataset was curated. The exact details of implementing their work are missing. The authors of the paper at https://ijrpr.com/uploads/V4ISSUE4/IJRPR12056.pdf claim they have used the TCIR dataset (which we have also cited in our work), but they also claim that their dataset is derived from INSAT 3D satellite, but the TCIR dataset has cited GridSat as their source which in turn uses the GOES satellite images thus contradicting their claims. Moreover, the lack of DOI for these papers also makes it impossible to cite them in our work. Apart from the two papers, we have found another paper indulging in comparable deficiencies in research integrity. For example, the authors of paper 10.1109/ICIDCA56705.2023.10099964  claim they have used cyclone images from INSAT 3D satellite from 2012-2021, but the INSAT-3D satellite was launched on 25 July 2013 thus suggesting false claims in their work. These deficiencies make these works unsuitable to be cited in our work. We assure you that our research is an independent and original contribution. We have meticulously developed a comprehensive dataset for cyclone detection and intensity estimation in popular formats used by the computer vision research community, our dataset is available on Zenodo at https://doi.org/10.5281/zenodo.8015544